# ACTIONS INSPIRE EVERY MOMENT: ONLINE ACTION-AUGMENTED DENSE VIDEO CAPTIONING

## ABSTRACT

Dense video captioning requires solving the challenging tasks of temporally localizing events and generating descriptive captions within long video sequences. Existing methods often struggle to capture the evolving context within video streams and to produce accurate temporal alignment. To address this, we propose an online retrieval-augmented approach that processes video segments incrementally while dynamically retrieving relevant action phrases from a pre-constructed action-text corpus. This enriches the contextual information for both the video representation and the subsequent text decoder, improving the caption generation. Additionally, we present image-based simulated video pretraining, which mitigates the reliance on extensive video datasets by using image-level text-paired data aligned with the online video captioning format. Our experiments on the ViTT, YouCook2, and ActivityNet benchmarks demonstrate that our model significantly outperforms both existing global and online methods, validating its effectiveness.

## 1 INTRODUCTION

As video content continues to grow exponentially, the need for automatic video understanding has become increasingly critical. Dense video captioning (Zhu et al., 2022; Wang et al., 2021a; Yang et al., 2023; Wu et al., 2024) involves generating detailed, temporally localized captions for multiple events or actions in long video sequences. This capability is crucial for various applications including video retrieval, summarization, and accessibility. Traditional video captioning approaches often generate a single, global caption for an entire video, missing the temporal granularity required to describe and localize specific events as they occur.

Recent work (Zhou et al., 2024) introduced the concept of online dense video captioning, where video segments are processed incrementally as they become available. This allows captions to be generated progressively, addressing the challenge of handling long video sequences by breaking them into manageable segments, thereby improving dense video captioning performance. Yet, even with this advancement, challenges remain in effectively capturing and utilizing the evolving context in video streams, which is critical for generating contextually relevant captions for each segment.

To address these challenges, we propose Actions Inspire Every Moment (AIEM), a novel online retrieval-augmented approach for dense video captioning. AIEM enhances dense video captioning by dynamically retrieving and integrating relevant action phrases from a pre-constructed action-text corpus as it processes video segments in an online manner. This integration enriches video representation and caption generation by providing the visual encoder and text decoder with evolving contextual information, resulting in more accurate and contextually relevant captions. Notably, we demonstrates that retrieving concise action phrases is more effective than using full captions, which can be redundant or less focused. Furthermore, to address the scarcity of large-scale dense video caption datasets, we explore an image-based simulated video pretraining, which leverages image-text paired data to align pretraining with the online video captioning format and improve model performance. Experiments on the ViTT, YouCook2, and ActivityNet benchmarks show significant performance gains, highlighting the effectiveness of our approach.

## 2 RELATED WORK

**Dense video captioning.** Dense video captioning is essential for video understanding, providing detailed descriptions of multiple events with their temporal localization. Unlike conventional

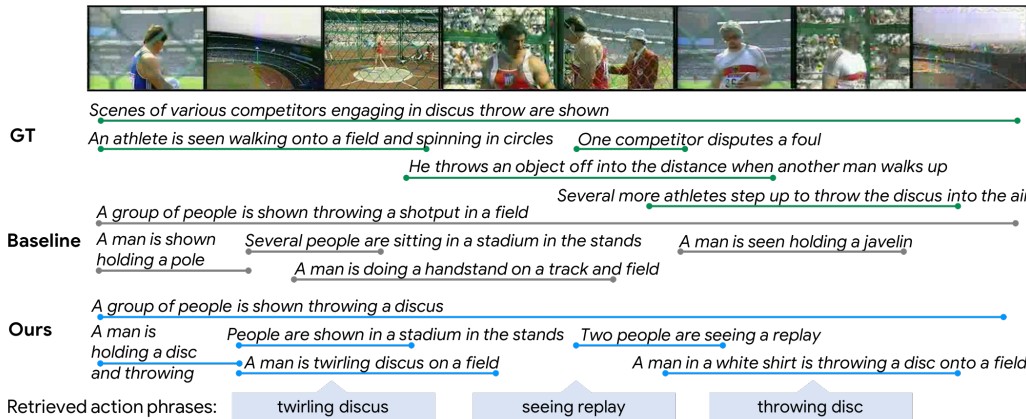

Figure 1: Dense video captioning results of our AIEM method. Ground truth captions and their timestamps (green), our non-augmented baseline (gray), and action-augmented model prediction (blue) with retrieved action phrases in blue boxes. Our model generates more accurate and temporally aligned captions, demonstrating the benefit of dynamic action retrieval and integration.

methods that produce a single embedding or caption per video (Wu & Krahenbuhl, 2021; Sun et al., 2022; Ashutosh et al., 2023; Gao et al., 2023; Cheng & Bertasius, 2022; Islam & Bertasius, 2022; Lin et al., 2022; Zhang et al., 2019; Ashutosh et al., 2023), dense video captioning offers temporally aligned captions, which is particularly beneficial for long, untrimmed videos in tasks such as retrieval, summarization, and accessibility. Approaches vary, with some using a two-stage method to detect segment boundaries before generating captions (Iashin & Rahtu, 2020), while others jointly predict boundaries and captions (Wang et al., 2018; 2021a; Zhang et al., 2022; Zala et al., 2023). Vid2Seq (Yang et al., 2023) introduces a sequence-based model, combining event captions and timestamps into a unified sequence for efficient processing. Recent multimodal and video LLMs (Lin et al., 2023; Song et al., 2024; Zhang et al., 2023; Li et al., 2023; Ren et al., 2024) typically process videos offline using short selected clips. While some of them are explored for dense video captioning, they often underperform compared to state-of-the-art methods. Most existing models typically rely on global, offline video processing, requiring access to the entire video and facing challenges with contextual relevance and temporal localization, especially in long videos. Our work focuses on online methods, with a particular emphasis on integrating online action retrieval to improve temporal alignment and captioning accuracy as the video is processed incrementally.

**Online dense video captioning.** Online video understanding, including tasks like online action detection (De Geest et al., 2016; Wang et al., 2021b; Kondratyuk et al., 2021; Zhao & Krähenbühl, 2022; Zhao et al., 2023) and online temporal action localization (Singh et al., 2017; Buch et al., 2017; Kang et al., 2021), focuses on predicting actions and their timing without access to future frames. This approach enables dynamic caption generation, making it more scalable for handling long and complex videos. Zhou et al. (2024) pioneered online dense video captioning demonstrating its benefits, while Chen et al. (2024) extended this idea to video-based dialogue. However, current online methods still face challenges in capturing dynamic context and recognizing intricate actions and events using only vision-based models. Our work introduces retrieval augmentation to incorporate timely and relevant external information about actions an events, improving contextual understanding and captioning accuracy in online settings.

**Retrieval augmented methods.** Retrieval-augmented methods, initially popular in NLP, have been widely used in vision-language tasks like video retrieval (Zhang et al., 2021; Jing et al., 2023; Chen et al., 2023), pretraining (Xu et al., 2021), and captioning (Xu et al., 2024; Kim et al., 2024). Most video captioning approaches rely on global, offline retrieval, where segment-level information is merged into a global context for captioning. However, this can limit temporal precision, as the global context may not align well with the dynamic nature of video content.

In this work, we propose a dynamic online retrieval mechanism coupled with autoregressive modeling, where action phrases are retrieved and integrated incrementally at each time step as the video streams. This autoregressive integration allows for a causal, segment-by-segment adaptation, im-

proving temporal alignment and caption quality by incorporating timely information from both current and previous segments By synchronizing the retrieval with the video's progression, our approach aims to improve upon the global retrieval methods. Unlike existing methods that use either text prefixing or embedding fusion, our approach integrates retrieved information as both a prefix to the text decoder and as embeddings fused with visual features. Additionally, We simplify action phrases into compact forms, such as action-object pairs, instead of using full captions as in other retrieval-augmented methods. This distills lengthy captions into compact action phrases, enhancing both efficiency and performance by focusing on essential information.

## 3 METHOD

### 3.1 PRELIMINARIES

**Captioning model.** Our captioning model follows the standard setup of a vision encoder followed by a text decoder. We utilize the CLIP model (Radford et al., 2021), pretrained on the LAION-2B (Schuhmann et al., 2021) dataset. Specifically, we adopt CLIP's Vision Transformer (ViT) as the visual encoder and the 12-layer Transformer as the text decoder. Since the original CLIP text model is designed for contrastive learning rather than text generation, we modify it by applying causal attention masking across all layers and further pretrain it for the image captioning task using the LAION-2B dataset. We keep the vision encoder frozen during the captioning training to preserve the original vision-text alignment from CLIP's pretraining.

**Dense video captioning.** Given a video $V \in \mathbb{R}^{T \times H \times W \times 3}$, our goal is to generate a set of temporally localized captions: $\{( [s_1][e_1][\text{caption text}_1] ), ..., ( [s_n] [e_n] [\text{caption text}_n] )\}$, where each caption is associated with start [s] and end [e] times that mark the event boundaries within the video. Inspired by Vid2Seq (Yang et al., 2023), we adopt their data format of representing temporal information as discrete vocabulary tokens. The start and end times [s] and [e] are included as text tokens within the caption sequence, enabling both temporal and caption information to be encoded in the text format. This avoids the need for separate output heads for time prediction and text generation.

**Autoregressive modeling for online video captioning.** Unlike global offline methods like Vid2Seq (Yang et al., 2023), which caption the entire video at once and become computationally expensive for long videos, Zhou et al. (2024) introduce an online approach where captions are generated incrementally as frames are processed. Building on this concept, our approach employs an autoregressive online model that processes video segment-by-segment, enhancing both temporal alignment and contextual coherence across current and past segments.

An overview of our model is illustrated in figure 2. We divide the video $V$ into $S$ segments process them incrementally, simulating an online scenario where segments become available over time. Each segment contains $L$ frames ($T = S \times L$) which are processed by the visual encoder, *e.g.* ViT which outputs $M$ tokens of dimension $D$ per segment. A token reduction Transformer then reduces the segment features by sampling $N$ tokens ($N \ll M$), resulting in a tensor of shape $N \times D$. As new segments are processed, let $S'$ denote the number of segments streamed up to the current time step, where $S' \leq S$. The outputs of all available segments are stacked as $S' \times N \times D$. On top of this, we apply an autoregressive Transformer with causal attention along the segment axis (S-axis). This incorporates information from both current and preceding segments, generating an evolving and contextualized video representation as segments are processed incrementally.

Once the autoregressive Transformer generates a contextualized video representation, the text decoder generates captions for each segment independently. Instead of concatenating all event captions into one long sequence, our method aligns captions with their respective video segments. During training, the target text for each segment is determined by the captions whose end times fall within the segment's temporal interval. This ensures that captions are accurately associated with the events from each segment. This allows each segment to include events that span multiple segments, with the start time [s] potentially from any prior segment and the end time [e] within the current segment (see figure 1). For a segment without events, we label it as "[BOS][EOS]" using Beginning/End-Of-Segment tokens. For a segment with multiple actions, captions are combined sequentially with corresponding start and end times as "[BOS][$s_1$][$e_1$][caption text$_1$][$s_2$][$e_2$][caption text$_2$] ... [EOS]". This segment-by-segment processing allows for the online generation of temporally localized captions as each segment is processed, avoiding the need to generate all captions at the end of the video. It also makes the approach efficient and well-suited for the online dense video captioning task.

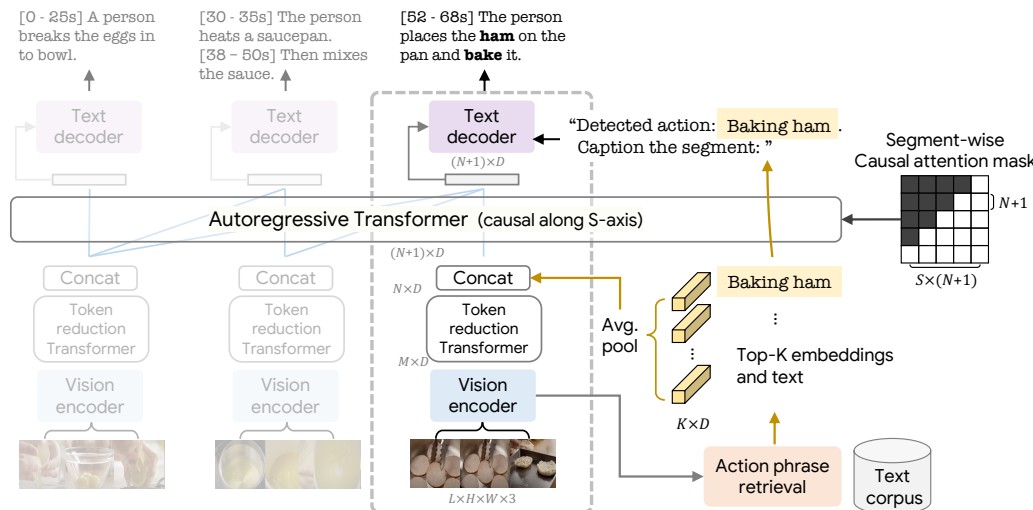

Figure 2: **Overview of AIEM**: Actions Inspire Every Moment framework for online action-augmented dense video captioning. Our model processes video segments incrementally, dynamically retrieving relevant action phrases from a pre-constructed text corpus. The top-k retrieved action phrases are integrated into both the visual encoder and text decoder for each segment, enhancing contextual information through fused embeddings and prefixing to the text decoder (shown in yellow).

## 3.2 ONLINE ACTION AUGMENTED DENSE VIDEO CAPTIONING

We present an approach that enhances online dense video captioning by dynamically retrieving and incorporating relevant action priors for each video segment. This provides additional context to the video representation and text decoder, improving the performance of dense video captioning.

**Action text corpus construction.** We first construct a corpus of action phrases to serve as contextual priors during the dense video captioning process. These action phrases are designed to capture the key actions or events in each video segment while also identifying the relevant objects

To construct this action phrase corpus, we collect captions from the training splits of existing video captioning datasets, such as ViTT, YouCook2, and ActivityNet. Importantly, we only utilize the text captions, excluding the corresponding video frames. This approach enables us to extend beyond video-text paired datasets and incorporate diverse action descriptions from text-based sources, allowing for a larger set of action phrases.

To explore a broader and less domain-specific corpus, we use the HowTo100M dataset (Miech et al., 2019), which contains instructional videos with transcribed speech subtitles. Since HowTo100M is not specifically designed for dense video captioning, it offers a more generic set of action descriptions. Again, we only use the subtitle text, discarding the video content. Our experiments show that this more generic corpus performs comparably well.

Unlike existing works that use raw video captions (Xu et al., 2024) which can be lengthy and less focused, we propose summarizing them into concise action phrases. We use a publicly available language model, Gemma (Team et al., 2024). It is prompted to extract key actions from raw captions, in the form of an action-object pair, *e.g.*, "baking ham". An example prompt is: *Your goal is to summarize the input sentence using as few words as possible. Focus on the words describing actions or events. Use singular nouns, avoid articles and numeric terms. Respond in the format of <action verb (ing)> <target object (if any)>. Input: {raw caption}. Answer:*

This approach ensures that the resulting action phrases are concise and focused, filtering out irrelevant details from the original captions. By summarizing captions in this way, we enhance the effectiveness of vision-text retrieval, leading to more accurate captioning results. Our experiments show that these simplified action phrases outperform the use of full captions. We collect 30,000 action phrases in total and precompute their text embeddings.

**Precomputation of text embeddings.** To optimize the retrieval of action phrases during training and inference, we precompute the text embeddings for the collected action phrases. This step is performed once at the precomputation stage, ensuring efficient retrieval without redundant embedding computations during the video captioning process. For this precomputation, we use the frozen instance of the CLIP text model, which retains its original contrastive learning capabilities, as opposed to the captioning-trained instance described in section 3.1. This frozen model is solely used for precomputing text embeddings and does not participate in the actual captioning process. The vision encoder is also kept frozen throughout the dense video captioning training. Once the text features are extracted from each action phrase, they are globally pooled into a single embedding.

**Online retrieval and integration of action phrases.** Our vision encoder processes each video segment $V_i \in \mathbb{R}^{L \times H \times W \times 3}$ in an online manner. After the visual features are extracted (before token reduction), global pooling is applied to represent the segment as a single embedding. The pooled segment embedding is then compared to the precomputed text embeddings of the action phrases using cosine similarity, selecting the top-k phrases based on their alignment. This ensures that the most relevant actions or events are identified for each video segment.

We integrate the retrieved action phrases into the captioning process in an online manner using several approaches. Note that each approach involves training a new model. **1) Prefixing the text decoder:** The retrieved action phrases are added as natural language prefixes to the text decoder. This approach offers direct guidance on the key actions or events in each video segment, improving caption accuracy and relevance by providing the decoder with explicit information about the segment's content. **2) Embedding fusion:** The text embeddings of the retrieved action phrases are fused with the visual embeddings after the token reduction and before the autoregressive Transformer (see section 3.1). The top-k text embeddings of the retrieved action phrases are average-pooled into one embedding, then concatenated with the token-reduced visual embeddings. This creates enriched multimodal features that offer a more comprehensive representation of each segment. The fused features are then processed by the autoregressive Transformer, enabling each segment to incorporate both visual and retrieved text information, along with the preceding sequences of such multimodal information. This results in a temporally and causally aware video representation, which is particularly important for dense video captioning, where understanding the temporal span and boundaries of events is critical. **3) Combined approach:** Lastly, we employ a hybrid method that combines both prefixing and embedding fusion. The action phrases serve as both natural language prefixes for the text decoder and fused embeddings with the visual features. This strategy leverages the strengths of both sources of information: it provides explicit action guidance for the decoder while enriching the autoregressive Transformer with a temporally-aligned, multimodal context. This combination enhances the model's ability to generate accurate and temporally aligned captions by maintaining awareness of immediate actions and the evolving context throughout the video stream.

**Mixed training.** We propose a mixed training strategy that alternates between action retrieval-augmented and standard (non-augmented) training. By randomly alternating between these modes during training, the model is better prepared for varied inference settings, ensuring consistent performance even without retrieval support.

**Frame construction for video segment representation.** To better capture the visual content within each video segment, we propose a frame sampling strategy that tiles multiple frames ($n^2$) into an $n \times n$ spatial grid. This approach better aligns with CLIP's pretraining on static images, enabling it to capture inter-frame relationships without video-specific adaptations. Empirically, it outperforms independently processing and merging frames with temporal positional embeddings of shape $L \times D$ added for each segment.

### 3.3 SIMULATED VIDEO PRETRAINING VIA IMAGE STITCHING

In addition, we explore an image-based pretraining method for dense video captioning to address the challenge of limited access to densely captioned video datasets. Unlike Vid2Seq (Yang et al., 2023), which uses large-scale video data with ASR-generated pseudo captions, we investigate whether we can benefit from image-based pretraining by simulating video sequences. We utilize the LAION-2B image-text paired dataset without relying on costly video datasets.

We simulate video sequences by stitching together multiple images, repeating each image for several frames. To create smoother transitions between images, we blend the pixels at the boundaries

and apply random augmentations to each frame, adding variability and avoiding overly monotonous sequences. Each stitched sequence is paired with its corresponding image captions, with the temporal span of each image defined by the stitching process. This setup replicates the dense video captioning task, reproducing the annotation format for temporally localized captions as described in section 3.1: $\{( [s_1][e_1][\text{caption text}_1] ), ..., ( [s_n] [e_n] [\text{caption text}_n] )\}$. By pretraining on these synthetic video sequences, our model learns to generate temporally localized captions from stitched image sequences. This process follows the same autoregressive framework for online dense video captioning (section 3.1), providing a warm start for the autoregressive Transformer and token reduction Transformer, which are newly added for dense video captioning.

## 4 EXPERIMENTAL RESULTS

**Datasets.** We evaluate our model on three widely-used dense video captioning datasets: ViTT (Huang et al., 2020), YouCook2 (Zhou et al., 2018a), and ActivityNet Captions (Heilbron et al., 2015). **ViTT** features instructional videos averaging 4.7 minutes in length with about 7 events per video, making it ideal for testing both caption generation and temporal alignment. **YouCook2**, consists of cooking videos averaging 5.3 minutes and 7.8 events per video, and **ActivityNet Captions** includes human activity videos with an average length of 2 minutes and 3.7 events per video.

**Evaluation metrics.** We use standard dense video captioning metrics for evaluation. SODA (Fujita et al., 2020) provides a comprehensive assessment of temporal alignment and caption accuracy across all event captions in a video. CIDEr (Vedantam et al., 2015) is averaged over IoU thresholds (0.3, 0.5, 0.7, 0.9) to measure caption relevance and alignment with ground truth events. We also report METEOR (Banerjee & Lavie, 2005) for caption quality, averaged over the same IoU thresholds, and the F1 score for temporal localization accuracy based on IoU with ground truth.

**Model architecture and training.** Our model consists of approximately 500M parameters, including 303M for the vision encoder (ViT-Large) and 128M for the text decoder, both initialized from the CLIP model pretrained on the publicly available LAION-2B dataset (Schuhmann et al., 2021). The text decoder is further pretrained for 0.2 epochs on the same LAION-2B dataset, focusing on the image captioning task (see section 3.1). For dense video captioning, we add the token reduction Transformer and autoregressive Transformer, each with 8 layers and 32M parameters. Unlike prior approaches that rely on large-scale video-text datasets for pretraining (Yang et al., 2023; Zhou et al., 2024) or specially curated datasets (Wu et al., 2024), our model only leverages image-text data for pretraining. To simulate video sequences, we sample 3 to 5 images from the LAION-2B dataset, repeating each image multiple times to form a 16-frame sequence. This simulated video data is then used to pretrain the entire dense video captioning model for 100,000 steps with batch size 32. This allows us to explore the effectiveness of image-based pretraining in scenarios where video-text paired datasets are limited or unavailable. During dense video captioning finetuning, the full model is trained for 20,000 steps with batch size 8, taking about 12 hours on 16 devices. The CLIP-initialized ViT encoder remains frozen throughout the pretraining and fine-tuning stages. At inference, we employ beam search to generate 6 candidate captions with a temperature of 1. Temporal non-maximum suppression (NMS) is applied to filter out intervals with a temporal IoU greater than 0.7, and the remaining intervals with their corresponding captions are used for evaluation.

**Video segment processing.** We divide each video into 16 segments ($S = 16$) and process them incrementally, simulating an online scenario where segments become available over time. Each segment consists of multiple frames. For our main comparison with state-of-the-art methods, we use a total of 144 frames per video, with 9 frames per segment ($L = 9$). Each frame is resized to 176×176 pixels, and the frames are then combined into a 3×3 grid, resulting in a 528×528 composite image per segment. For the ablation studies, we use a simpler setup, with 16 frames per video and 1 frame per segment ($L = 1$), resizing each frame to 256×256 pixels. The token reduction Transformer samples $N = 64$ tokens from each segment.

### 4.1 ESTABLISHING A BASELINE MODEL

We present our baseline model for online dense video captioning and compare it with a global, offline counterpart. Although our approach shares the basic principle of online processing with the recent method (Zhou et al., 2024), we independently establish our own baseline, as their model

weights are not publicly available. Both methods utilize CLIP pretrained models, particularly the CLIP vision encoder ViT-Large, but differ in infrastructure and model design. Zhou et al. (2024) use a larger 256M parameter T5-Base model (Raffel et al., 2020) as the text decoder, pretrained on Web text corpora. In contrast, we explore a smaller 128M parameter CLIP text model, further trained on the LAION-2B dataset for image captioning. Despite its smaller size, our experiments show that it remains highly effective. Their method also introduces additional features, such as a recursive feedback loop, which we omit as it did not provide gains in our setup, along with a token clustering-based memory module and an online decoding algorithm, which we also did not use. Instead, as described in section 3.1, our model processes videos by segments, using an inter-segment autoregressive Transformer followed by a factorized per-segment text decoder, resulting in a distinct yet effective online model.

We perform an exploratory comparison between our online model and its global captioning counterpart to assess the performance of our baseline for online dense video captioning. For the global baseline, we modified our online model by removing the segment-wise processing. The global model processes all video segments simultaneously, assuming access to the entire video. In this setup, the text decoder operates on the full set of segment representations at once, generating a single long caption sequence that concatenates all event captions and their timestamps, similar to the global method (Yang et al., 2023). The key difference between the two models lies in how the text decoder processes video segments: segment-by-segment for the online model versus all-at-once for the global model. We evaluate both baselines on the ViTT benchmark as follows:

| method | SODA | CIDEr | METEOR |
|---|---|---|---|
| Global (offline) | 6.8 | 23.1 | 6.5 |
| Online | 7.6 | 27.7 | 7.2 |

The results show that our online model outperforms the global model across all metrics. This is likely due to the model's ability to process video segments independently, leading to more temporally accurate captions. Also, our global and online baselines are in a comparable range to the state-of-the-art results of Vid2Seq (Yang et al., 2023) and Streaming (Zhou et al., 2024), respectively (see Table 9). This provides a reasonable foundation for evaluating the impact of our proposed contributions in the following sections, without introducing significant inherent advantages or disadvantages. Building on the strengths of the online per-segment model, our main contribution in this work is to further enhance dense video captioning through the incorporation of online action retrieval augmentation, dynamically enriching the representation of each video segment as the video is streamed.

## 4.2 ABLATION STUDIES

Our ablation uses the ViTT dataset (Tables 1-8). We use a simplified setup where each video is divided into 16 segments, with 1 frame per segment at 256×256 resolution, unless otherwise noted.

### 4.2.1 ONLINE ACTION RETRIEVAL AND INTEGRATION

**Retrieval integration format.** We compare different methods for incorporating the retrieved action phrases into our online dense video captioning model (see section 3.2). Table 1 compares the performance of no retrieval, prefixing the text decoder with action phrases, text embedding fusion with vision features, and the combined approach. The combined approach (our full method), which integrates action phrases both as text prefixes and fused multimodal embeddings, performs best across all metrics, highlighting the complementary benefits of both strategies.

**Number of retrieved phrases.** Next, we study the effect of number of retrieved action phrases by varying the top-k phrases used as prefixes and embeddings. As presented in Table 2, using 1 prefix and 10 embeddings offers the best balance, resulting in the highest scores across all metrics.

**Action text corpus.** Table 3 compares various sources for action phrases, including original captions and summarized action phrases from YouCook2, ViTT, and ActivityNet Captions, as well as 12k object names from the V3Det (Wang et al., 2023) and 700 action names from the Kinetics (Carreira et al., 2018; 2019) dataset. The summarized action phrases significantly outperform full captions, highlighting the importance of concise action descriptors. The best performance is achieved with summarized phrases from the union of YouCook2, ViTT, and ActivityNet. Notably, HowTo100M summarized corpus performs comparably well despite being more generic, showing the robustness of our approach with generic action descriptions. Table 4 further shows that ViTT

| method | S | C | M |
|---|---|---|---|
| No retrieval | 7.6 | 27.7 | 7.2 |
| Prefixing text decoder | 8.5 | 31.4 | 8.3 |
| Embedding fusion | 9.2 | 35.6 | 9.0 |
| Both combined | 9.9 | 37.2 | 9.6 |

Table 1: **Retrieval integration.** Combining prefixing and embedding fusion performs best.

| k (prefix) | k (embeds) | S | C | M |
|---|---|---|---|---|
| 1 prefix | 1 embed | 8.7 | 32.5 | 8.4 |
| 1 prefix | 10 embeds | 9.9 | 37.2 | 9.6 |
| 1 prefix | 50 embeds | 9.8 | 37.0 | 9.4 |
| 5 prefixes | 1 embeds | 8.1 | 30.0 | 7.8 |
| 5 prefixes | 10 embed | 8.3 | 30.8 | 8.0 |

Table 2: **Number of retrieved action phrases** used as prefixes and fused embeddings

| method | S | C | M |
|---|---|---|---|
| Y+V+A original captions | 8.4 | 33.7 | 8.1 |
| Y+V+A action phrases | 9.9 | 37.2 | 9.6 |
| HowTo100M action phrases | 9.9 | 37.0 | 9.7 |
| V3Det object names | 9.0 | 34.9 | 8.6 |
| Kinetics action names | 9.1 | 35.4 | 8.9 |

Table 3: **Action text corpus.** Summarized action phrases from the union of YouCook2 (Y), ViTT (V), ActivityNet Captions (A) as well as the HowTo100M perform the best.

| method | S | C | M |
|---|---|---|---|
| ViTT action phrases | 9.7 | 36.9 | 9.3 |
| Y+A action phrases | 9.5 | 36.3 | 9.0 |

Table 4: **Action text corpus** (more analysis).

| method | S | C | M |
|---|---|---|---|
| Global (offline, all concat) | 8.4 | 29.5 | 7.8 |
| Online (per-segment) | 9.9 | 37.2 | 9.6 |

Table 5: **Online retrieval and integration** significantly outperforms global, offline retrieval.

| method | Inference with retrieval | | | Inference without retrieval | | |
|---|---|---|---|---|---|---|
| | S | C | M | S | C | M |
| Non-augmented training | N/A | N/A | N/A | 7.6 | 27.7 | 7.2 |
| Action-augmented training | 9.9 | 37.2 | 9.6 | 4.2 | 18.9 | 3.8 |
| Mixed training | 9.8 | 37.4 | 9.4 | 7.8 | 28.0 | 7.5 |

Table 6: **Mixed training** enhances the model's adaptability, improving performance in scenarios both with and without retrieval-based augmentation.

action phrases yield better results than the out-of-domain YouCook2 + ActivityNet (Y+A) corpus, likely due to closer alignment with the train/test data. Nevertheless, Y+A still achieves competitive results, indicating our method's generalization ability.

**Online vs global retrieval.**   To evaluate the benefits of online retrieval over global retrieval, we compare the two approaches in Table 5. In the global retrieval setup, action phrases are retrieved for the entire video and concatenated both as prefixes and embeddings. The results show that online retrieval significantly outperforms global retrieval. This suggests that the online retrieval provides more temporally relevant and localized information for each segment, leading to improved performance in online dense video captioning.

**Mixed training with non-augmented setting.**   Our mixed training strategy alternates between action-augmented and non-augmented training to enhance the model's adaptability to inference settings without retrieval-based augmentation. Table 6 shows the mixed training significantly improves the non-augmented inference while maintaining strong performance when augmentation is available.

**Frame construction per video segment.**   We evaluate different strategies for constructing video segment representations, as shown in Table 7. The $3 \times 3$ tiled combination of 9 frames achieves the best results across all metrics. This approach aligns well with CLIP's pretraining on static images especially given our frozen vision encoder (see section 3.1), capturing inter-frame information more effectively and leading to improved captioning performance compared to using single frames or processing multiple frames independently.

### 4.2.2   Image-based simulated video pretraining

Table 8 evaluates the effect of image-based simulated video pretraining (section 3.3) which simulates video sequences by stitching together multiple images from the LAION-2B image-text dataset. This provides notable improvements, with an effective warm-start for our dense video captioning model.

### 4.3   Comparison to State-of-the-art Methods

To compare with state-of-the-art models, our full method uses 144 frames per video with 9 frames per segment, along with online action augmentation and image-based simulated video pretraining.

As shown in Table 9, our model outperforms both global and online methods across all benchmarks. On the ViTT dataset, it achieves 11.8 SODA, 44.9 CIDEr, 11.3 METEOR, and 45.1 F1, surpassing

| # frames per segment | S | C | M |
|---|---|---|---|
| 1 frame (1×256×256) | 9.8 | 37.4 | 9.4 |
| 9 frames (9×256×256) | 10.5 | 40.5 | 10.0 |
| 3×3 tiled 9 frames (1×528×528) | 11.1 | 43.8 | 10.4 |

| method | S | C | M |
|---|---|---|---|
| w/o simulated video pretraining | 9.8 | 37.4 | 9.4 |
| Simulated video pretraining | 10.8 | 39.1 | 10.3 |

Table 7: **Frame construction per video segment.**  Table 8: **Image-based pretraining.**

| method | backbone | ViTT | | | | YouCook2 | | | | ActivityNet | | | |
|---|---|---|---|---|---|---|---|---|---|---|---|---|---|
| | | S | C | M | F1 | S | C | M | F1 | S | C | M | F1 |
| E2ESG (Zhu et al., 2022) | C3D | - | - | - | - | - | 25.0 | 3.5 | - | - | - | - | - |
| MT (Zhou et al., 2018b) | TSN | - | - | - | - | - | 6.1 | 3.2 | - | - | 9.3 | 5.0 | - |
| PDVC (Wang et al., 2021a) | TSN | - | - | - | - | 4.9 | 28.9 | 5.7 | - | 6.0 | 29.3 | 7.6 | - |
| GIT (Wang et al., 2022) | GIT | 7.1 | 15.1 | 3.4 | 32.5 | 3.1 | 12.1 | 3.4 | 17.7 | 5.7 | 29.8 | 7.8 | 50.6 |
| OmniViD (Wang et al., 2024) | VideoSwin | - | - | - | - | - | - | - | - | - | 26.0 | 7.5 | - |
| TimeChat (Ren et al., 2024) | 7B MLLM | - | - | - | - | 3.4 | 11.0 | - | 19.5 | - | - | - | - |
| Vid2Seq † (Yang et al., 2023) | CLIP | 9.8 | 23.0 | 5.0 | 37.7 | 5.7 | 25.3 | 6.4 | 23.5 | 5.9 | 30.2 | 8.5 | 51.8 |
| DoYou (Kim et al., 2024) | CLIP | - | - | - | - | 5.3 | 31.7 | 6.1 | 33.4 | 6.2 | 33.0 | 8.6 | 55.2 |
| DIBS (Wu et al., 2024) | CLIP | - | - | - | - | 6.4 | 44.4 | 7.5 | 31.4 | 5.9 | 31.9 | 8.9 | 55.6 |
| Streaming ⋆ (Zhou et al., 2024) | CLIP | 10.0 | 25.2 | 5.8 | 35.4 | 6.0 | 32.9 | 7.1 | 24.1 | 6.2 | 37.8 | 10.0 | 52.9 |
| AIEM (our full method) ⋆ | CLIP | 11.8 | 44.9 | 11.3 | 45.1 | 8.7 | 48.5 | 10.6 | 34.8 | 8.2 | 38.5 | 14.0 | 55.8 |

Table 9: **Comparison to the state-of-the-art on dense video captioning.** We evaluate on the ViTT, YouCook2, and ActivityNet benchmarks. We report SODA (S), CIDEr (C), and METEOR (M) for caption quality, and F1 score for temporal localization. Our full method uses 144 frames per video with 9 frames per segment. †: version with visual-only inputs. ⋆: online methods (all others are global offline methods).

the previous best method Streaming (Zhou et al., 2024) across all metrics, including gains of +1.8 SODA scores. Similar improvements are observed on YouCook2 and ActivityNet across all metrics, including gains of +2.3 and +2.0 in SODA, over the previous best methods.

These results demonstrate the effectiveness of our online action-augmented dense video captioning approach. Our dynamic action retrieval and integration provide a significant advantage in tasks requiring fine-grained temporal localization and action description, while avoiding the need for access to the entire video as required by global methods. Unlike previous methods that rely on large-scale video-text datasets like YT-Temporal-1B (Zellers et al., 2022) for pretraining (Yang et al., 2023; Zhou et al., 2024) or specifically curated HowTo100M video-pretraining datasets (Wu et al., 2024), our model leverages only image-text data for pretraining. We employ an online retrieval-augmented strategy combined with image-based pretraining, delivering strong performance without the need for extensive video-text pairs. Notably, our retrieval of concise action phrases proves more effective than using full captions, as it avoids redundancy and is more readily available.

While additional performance gains could come from incorporating Automatic Speech Recognition (ASR) (Yang et al., 2023; Wang et al., 2021a), we intentionally avoid it. ASR often overlaps with ground truth captions and is closely tied to action occurrences, which can potentially inflate performance metrics without accurately reflecting the model's visual understanding. Although some earlier models use ASR to make the task easier, more recent methods tend to avoid it.

## 4.4 VISUALIZATION

Figure 1 presents the results of our method on the ActivityNet Captions dataset. Our model produces more temporally aligned and accurate captions, such as identifying actions like 'twirling discus' and 'throwing disc', demonstrating the effectiveness of dynamic action retrieval and integration.

## 5 CONCLUSION

In this work, we introduced Actions Inspire Every Moment (AIEM), a novel approach for online dense video captioning that enhances caption quality by dynamically retrieving and integrating relevant action phrases at each time step. Our method leverages an autoregressive model to align retrieval with the video's temporal progression, enabling more precise and contextually appropriate captions. Additionally, our image-based simulated video pretraining further improves performance. Experiments on the ViTT, YouCook2, and ActivityNet benchmarks demonstrate that AIEM significantly improves both caption quality and temporal localization, outperforming state-of-the-art global and online methods, establishing it as a leading approach in the field.

## ETHICS STATEMENT AND LIMITATIONS

The task of dense video captioning inherently involves a level of subjectivity, which can lead to ambiguous boundaries and mismatches between model outputs and the ground truth data used for evaluation. This may result in discrepancies between the model's performance and its evaluation through standard metrics. To construct our action text corpus, we utilize a publicly available language model for retrieval augmentation. While these models perform well on benchmarks, they may carry biases, stereotypes, or inaccuracies from the data they were trained on. Since our corpus is generated and filtered automatically, there may be instances where undesirable content persists. While we made efforts to mitigate these issues by enforcing strict formatting in the model's output, it remains essential to re-assess these models before applying them for specific purposes. Our model is designed for research purposes with the primary goal of evaluating its performance relative to state-of-the-art methods. While it introduces new capabilities that could inspire further positive advancements in research, the model is not intended for commercial or non-research applications.

## REPRODUCIBILITY STATEMENT

Our method is based on the CLIP model pretrained on the publicly available LAION-2B dataset (Schuhmann et al., 2021). Our further pretraining methods also use the same dataset. We provide all details on model architecture, pretraining, finetuning, and inference in section 4 and section A.1 of the Appendix. For action text corpus construction, we use the publicly available Gemma model (Team et al., 2024) alongside the dense video captioning datasets or HowTo100M (Miech et al., 2019) dataset, all of which are publicly available. Details of the corpus construction are provided in section 3.2 and section A.3 of the Appendix. The dense video captioning benchmarks used in this work, including ViTT (Huang et al., 2020), YouCook2 (Zhou et al., 2018a), and ActivityNet (Heilbron et al., 2015), are publicly available and widely used in the community. All evaluation settings follow established practices.

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

# A APPENDIX

## A.1 ADDITIONAL IMPLEMENTATION DETAILS

We utilize the CLIP model pretrained on the LAION-2B dataset. Specifically, we use its ViT-Large model (303M parameters) and its 12-layer Transformer text model (128M parameters). The CLIP-initialized ViT is kept frozen throughout all stages of training described below.

We further pretrain the CLIP text model on the image captioning task using the same LAION-2B dataset, with batch size 1024 for 0.2 epochs. We use the Adam optimizer with momentum 0.9, an initial learning rate (LR) of 5e-5, 5000 warmup steps, linear LR decay, weight decay 1e-2.

For dense video captioning, the token reduction Transformer and autoregressive Transformer modules are added. Each module consists of 8 layers with a model dimension of 512, and 32 M parameters. In total, the entire model contains approximately 500M parameters.

As described in section 3.3, we apply image-based simulated video pretraining on the entire model, including the newly added modules. To simulate video sequences, we sample 3 to 5 images from the LAION-2B dataset, repeating each image multiple times to form a 16-frame sequence. To create smoother transitions, we blend pixels at the boundaries by applying a weighted sum of two images, using a randomly selected blending ratio $\alpha \in [0.1, 0.9]$, *e.g.*, blending pixels of images A and B as $\alpha A + (1 - \alpha)B$. We apply random augmentations to each frame to avoid overly monotonous sequences. This pretraining follows the same segment-by-segment autoregressive framework for online dense video captioning. We use a batch size of 32 and train the model for 100000 steps. The optimizer is Adam with momentum 0.9, an initial LR of 1e-4, 5000 warmup steps, cosine LR decay, and a weight decay of 1e-5.

When finetuning on dense video captioning, the model is trained for 20000 steps with a batch size 16. We again use the Adam optimizer with momentum 0.9, an initial LR of 1e-4, 5000 warmup steps, cosine LR decay and a weight decay of 1e-5. For time tokenization, we use relative time tokens following Vid2Seq (Yang et al., 2023). We quantize a video of duration $T$ frames into $B = 32$ equally spaced time bins.

For inference, we follow the standard protocol to use beam search, with a beam size of 6 and temperature 1, followed by temporal NMS with a threshold of 0.7 to remove overlapping intervals.

## A.2 COMPUTATIONAL COST OF OUR MODEL

Our model uses 410 GFLOPs per segment, totaling 6560 GFLOPs for 16 segments. The retrieval operates on the precomputed text embeddings.

## A.3 PROMPTING FOR ACTION TEXT CORPUS CONSTRUCTION

We construct a corpus of action phrases to serve as contextual priors during the dense video captioning process. These phrases are designed to capture key actions or events in each video segment and identify relevant objects. To build this corpus, we draw from two main sources: 1) Captions from the training splits of dense video captioning datasets: ViTT, YouCook2, and ActivityNet, where we use only the text captions excluding the video frames. 2) A broader, less domain-specific corpus from the HowTo100M dataset (Miech et al., 2019), again using only the subtitle text without the video content.

Unlike previous methods that use raw video captions, which tend to be lengthy and unfocused (Xu et al., 2024), we summarize these captions into concise action phrases using the publicly available language model, Gemma (Team et al. (2024), huggingface.co/google/gemma-2-27b).

To improve the extraction process, we refine the prompt to ensure concise and consistently formatted action phrases. Specifically, we emphasize singular nouns, avoid numerical terms, and enforce a strict format of <action verb(ing)> <target object (if any)>. For example, our prompt is: *Your goal is to summarize the input sentence using as few words as possible. Focus on the words describing actions or events. Use singular nouns, avoid articles and numeric terms. Respond in the format of <action verb (ing)> <target object (if any)>. Input: {raw caption}. Answer:*

For HowTo100M subtitles, which are often longer, we adjust the prompt to focus on extracting a single main action or event: *The input is video subtitle text. Choose the main action or event in the video and summarize it using as few words as possible. Focus on the words describing actions or events. Use singular nouns, avoid articles and numeric terms. Respond in the format of <action verb(ing)> <target object (if any)>. Input: {video subtitles}. Answer:*

These prompts effectively generate concise action phrases. After processing all text in each source, we deduplicate phrases by merging those with the same set of words, regardless of word order. After filtering out some least frequent phrases, we obtain 30,000 action phrases for each corpus.

## A.4 ADDITIONAL ABLATIONS

We conduct additional ablation studies using the same setup described in section 4.2, where we use 16 frames per video, with 1 frame per segment at a resolution of 256×256 pixels, and report the

| text decoder pretraining | S | C | M |
|---|---|---|---|
| ✓ | 9.9 | 37.2 | 9.6 |
| x | 8.3 | 30.4 | 8.0 |

Table 10: **Ablation on text model pretraining** on the image captioning task. Results on ViTT.

| size of action text corpus | S | C | M |
|---|---|---|---|
| 1% | 8.7 | 33.0 | 8.4 |
| 10% | 9.3 | 35.2 | 9.1 |
| 50% | 9.7 | 36.7 | 9.5 |
| 100% | 9.9 | 37.2 | 9.6 |

Table 11: **Effect of size of action text corpus.** Results on ViTT.

| # segments | # frames | S | C | M |
|---|---|---|---|---|
| 8 | 8 | 9.3 | 36.2 | 8.9 |
| 8 | 16 | 9.6 | 37.0 | 9.1 |
| 16 | 16 | 9.9 | 37.2 | 9.6 |
| 16 | 32 | 10.0 | 37.7 | 9.8 |
| 32 | 32 | 9.7 | 36.8 | 9.0 |

Table 12: **Number of segments** which controls the number of decoding outputs. Results on ViTT.

| method | ViTT | | | | YouCook2 | | | | ActivityNet | | | |
|---|---|---|---|---|---|---|---|---|---|---|---|---|
| | S | C | M | F1 | S | C | M | F1 | S | C | M | F1 |
| baseline online model | 7.6 | 27.7 | 7.2 | 34.0 | 5.3 | 27.7 | 6.8 | 22.4 | 5.4 | 31.6 | 9.8 | 45.8 |
| online action-augmentation | 9.8 | 37.4 | 9.4 | 35.5 | 6.9 | 39.4 | 8.0 | 25.5 | 6.7 | 34.6 | 11.2 | 46.2 |
| image-based simulated video pretraining | 8.9 | 34.1 | 8.5 | 37.8 | 6.1 | 35.0 | 7.2 | 27.8 | 6.2 | 33.7 | 10.4 | 47.6 |
| both combined | 10.8 | 39.1 | 10.3 | 39.2 | 8.0 | 45.6 | 9.3 | 30.7 | 7.5 | 36.4 | 12.1 | 49.9 |

Table 13: **Ablation of our method on ViTT, YouCook2, ActivityNet datasets.** This ablation uses $S$=16 segments per video, $L$=1 frame per segment at a resolution of $256 \times 256$ pixels.

| method | online | video-text pretraining | backbone |
|---|---|---|---|
| E2ESG (Zhu et al., 2022) | N | ∅ | C3D |
| PDVC (Wang et al., 2021a) | N | ∅ | TSN |
| OmniViD (Wang et al., 2024) | N | Kinetics | VideoSwin + Bart |
| TimeChat (Ren et al., 2024) | N | YT-Temporal, ViTT, ActivityNet, etc. | Eva-CLIP-G + Llama-7B |
| Vid2Seq † (Yang et al., 2023) | N | YT-Temporal-1B | CLIP-L + Bert-B |
| DoYou (Kim et al., 2024) | N | ∅ | CLIP-L |
| DIBS (Wu et al., 2024) | N | Howto100M | CLIP-L |
| Streaming (Zhou et al., 2024) | Y | YT-Temporal-1B | CLIP-L + Bert-B |
| AIEM (ours) | Y | ∅ | CLIP-L |

Table 14: **Comparison to the state-of-the-art on dense video captioning.**

results on the ViTT dataset. In this ablation, the mixed training (Table 6) and image-based simulated video pretraining (Table 8) are *not* used, unless otherwise noted.

**Effect of text decoder pretraining.** In Table 10, we show the effect of pretraining the text decoder for image captioning using the LAION-2B dataset (section 3.1). While image captioning pretraining improves performance, our model still performs reasonably well without it. Notably, most recent dense video captioning methods (Yang et al., 2023; Wang et al., 2024; Ren et al., 2024; Wu et al., 2024; Zhou et al., 2024) employ language pretraining for their text decoders, and we follow this approach to enhance the captioning performance.

**Ablation on the size of action text corpus.** Table 11 presents the effect of varying the size of the action text corpus. We randomly sample subsets of the corpus at 1%, 10%, 50%, and 100%. Increasing the corpus size improves performance, with the most notable gains observed up to 50%. This shows that our constructed corpus is effective in covering a broad range of action phrases for retrieval augmentation.

**Ablation on the number of segments.** Table 12 studies the effect of the number of segments. Overall, more frames improves performance. Across different segment configurations, our model performs robustly overall, with 16 segments yielding the best results.

**Ablation on ViTT, YouCook2, ActivityNet datasets.** In Table 13, we evaluate the effects of our key method components. The baseline refers to the online model described in section 4.1.

We observe both our main contributions – online action-augmentation section 3.2 and image-based simulated video pretraining section 3.3 – make complimentary improvements to performance across all three benchmarks: ViTT, YouCook2, and ActivityNet.

## A.5    COMPARISON OF APPROACHES IN EXISTING METHODS

Table 14 compares various strategies used in existing methods, focusing on key aspects such as support for online video captioning, reliance on video-text pretraining, and the backbone models employed.

