# OpenReview forum: "Actions Inspire Every Moment: Online Action-Augmented Dense Video Captioning"
_ICLR.cc/2025/Conference — ICLR 2025 Conference Withdrawn Submission_

### Official Review · Reviewer_UpTg · 2024-10-30

**Soundness:** 3
**Presentation:** 3
**Contribution:** 3
**Rating:** 6
**Confidence:** 3

**Summary:**

This paper proposes an online retrieval-augmented dense video captioning approach that utilizes action phrases with text assistance. Additionally, the authors introduce image-based simulated video pretraining, which reduces reliance on extensive video datasets by using image-level text-paired data aligned with the online video captioning format. They conducted experiments on the ViTT, YouCook2, and ActivityNet benchmarks to validate its effectiveness.

**Strengths:**

1. This work focuses on dense video captioning in video streaming setting. The problem to be studied is explained very clearly.
2. The figures are informative and the tables are well-organized.
3. The ablation experiments effectively validate the effectiveness of online action retrieval and integration and image-based simulated video pretraining.
4. The authors conducted comparisons with existing dense video caption models across multiple benchmarks, validating the effectiveness of their proposed model.

**Weaknesses:**

1. The authors need to provide a clearer explanation of the segmentation strategy. If a sequence is divided into different segments, does that imply it represents two distinct actions? This seems rather counterintuitive.
2. The authors should scale their validation methods to longer video sequences. The motivation of the paper is to address evolving contexts in video streams through an online strategy. Therefore, using longer video lengths, such as those found in ego datasets ego4d[1],egoSchema[2] ranging from dozens of minutes to hours, would better validate the effectiveness of their approach.

[1] Ego4D: Around the World in 3,000 Hours of Egocentric Video. https://arxiv.org/pdf/2110.07058
[2] EgoSchema: A Diagnostic Benchmark for Very Long-form Video Language Understanding. https://egoschema.github.io

**Questions:**

1. Could the authors provide a comparison of inference times in the tables, considering that both retrieval and streaming output are time-consuming processes, to demonstrate the model's performance in practical use cases?
2. Could you further elaborate on the segmentation strategy, particularly regarding cases where the same sequence is divided into different segments?

---

### Official Review · Reviewer_pWav · 2024-11-03

**Soundness:** 3
**Presentation:** 2
**Contribution:** 3
**Rating:** 5
**Confidence:** 4

**Summary:**

This submission focuses on the dense video captioning problem. The authors propose an online retrieval-augmented method to better handle the challenges of event localization and dense descriptions of long videos. The model incrementally processes pre-segmented video segments and dynamically retrieves relevant action phrases from a pre-built action-to-text corpus. During inference, the authors use the Transformer model and infer at the end of each pre-defined segment to generate dense descriptions. Experimental results on multiple benchmarks and ablation studies show that the proposed method outperforms existing methods, validating its effectiveness.

**Strengths:**

1. The proposed model and generation strategies are meaningful. I would say that authors really think about the tasks and methods are well-aligned with the motivation of solve this online dense video captioning tasks.
2. The idea to use the LLM to process original texts into action phrases is also great. It not only cleans the texts, but also helps to improve the retreival performance.

**Weaknesses:**

In my options, some motivations and choices in this submission are kind of arbitrary, and not well-supported by experiments and analysis.

1. I think one really interesting way to evaluate the proposed method is to show its performance in open-vocabulary/open-world/zero-shot dense video captioning settings. The authors mention some similar ideas in lines 305-306: "This allows us to explore the effectiveness of image-based pretraining in scenarios where video-text paired datasets are limited or unavailable." However, I think a stronger point is how the model can significantly improve in these settings with Top-K retrieval. Reference evaluation settings can be found in papers like [1, 2, 3]. One simple approach would be to perform a quick analysis of which kinds of actions typically pose challenges for baseline models, and whether these actions are successfully retrieved and generated by using the Top-K module. I just suggested a quick evaluation method here, but the authors could further develop this idea, which I believe would enhance the novelty of this work by going beyond conventional benchmarks. It would be great to see any relevant evaluation, and I would be happy to raise my score with this relevant analysis.

2. The videos have to be pre-segmented into different numbers of segments in order to be fed into the model. From my intuition, I think the best approach would be to make the ending point of segmentations a learnable choice. But I do understand that it may be challenging to design and learn in that case. I see that the authors perform some ablation studies in Table 12 to examine the choice of the number of segments and frames. I think some statistical information like how many captions are fully covered in single segments/the distribution of how many segements a caption usually be splitted into, could be more helpful for readers to understand this model design.

References:

[1] Retrieval Enhanced Zero-Shot Video Captioning, Ma et al.

[2] Learning to Compose Topic-Aware Mixture of Experts for Zero-Shot Video Captioning, Wang et al, AAAI 2019.

[3] Zero-Shot Video Captioning by Evolving Pseudo-tokens, Tewel et al, BMVC 2023.

**Questions:**

Most of my concerns have been put in the weakness section. Here I list some minor points:

1. In all tables, it would be easier to read by highlighting or bolding the best results.
2. The authors use the mixed metric SODA to evaluate the model performance. Since the model can generate the time stamp of each captions [Xs - Ys], it would be helpful to directly quantitively evaluate the dense temporal alignment.
3. It would be better to provide more case studies, e.g. in the Appendix section. Some error analysis would be also appreciated, such as in what cases, the model still cannot generate poor captions. And further questions could include whether poor cpations come from the language model, the text corpus, retrieval failures, or other factors.

---

### Official Review · Reviewer_ML2Z · 2024-11-04

**Soundness:** 2
**Presentation:** 2
**Contribution:** 2
**Rating:** 3
**Confidence:** 4

**Summary:**

This paper proposes a framework which integrates the idea from streaming dense video captioning and retrieval-augmented generation. Experiments on serveral benchmarks illustrate its effectiveness.

**Strengths:**

Extensive experiments illustrate that the proposed framework is an effective composition of the core ideas from streaming dense video captioning and retrieval-augmented caption generation.

**Weaknesses:**

1. Limited novelty: The primary contributions of this paper are heavily based on recent works$^{[1, 2]}$, with no significant improvements over previous methods. The use of retrieval-augmented generation is a common technique, and its application here adds little to the research community. The utilization of action information for video captioning has also been clained in previous works $^{[3]}$.

2. Lack of in-depth analysis:  While this paper highlights the importance of information in verbs/actions compared to previous retrieval-augmented methods, it overlooks the potential drawbacks. Many actions can have similar visual representations, and relying on retrieved action information without correction methods may lead to errors. It would be beneficial to conduct a deeper analysis of the impact of incorporating action information, such as examining the accuracy of action recall and the distribution of actions in your dataset. And if the recall rate is high, indicating precise action perception by the model, what advantages does retrieval-augmentation offer by introducing more relevant information? Additionally, relying on actions seems to complicate the definition of learning objectives: “[BOS][s1][e1][caption text1][s2][e2][caption text2] ... [EOS]”. For a common phrase like “Pick up the apple and peel it,” what should the corresponding statement be?

3. Poor presentation: Section 3 includes too many design details, presented mostly in text without considering the reader's experience. Consider enhancing this by incorporating illustrations and reducing the amount of text. It's also crucial to focus on simplicity and accuracy in your presentation.


[1] Streaming Dense Video Captioning. CVPR 2024.

[2] Do You Remember? Dense Video Captioning with Cross-Modal Memory Retrieval. CVPR 2024.

[3] Syntax-Aware Action Targeting for Video Captioning. CVPR 2020.

**Questions:**

As in weaknesses.

---

### Official Review · Reviewer_KcsL · 2024-11-04

**Soundness:** 3
**Presentation:** 3
**Contribution:** 2
**Rating:** 5
**Confidence:** 4

**Summary:**

This paper introduces an online dense video captioning approach that dynamically retrieves and integrates relevant action phrases processing video segments incrementally, combining both text prefix and embedding fusion strategies. In addition, the paper proposes an image-based video pretraining method that reduces reliance on video datasets while achieving SOTA performance across multiple benchmarks.

**Strengths:**

- The paper introduces a novel online retrieval-augmented approach for dense video captioning.
- The use of dynamic action phrase retrieval and integration is well-motivated.
- The image-based simulated video pretraining approach reduces reliance on video datasets.

**Weaknesses:**

* The paper lacks a strong theoretical justification for why action phrase retrieval works better in comparison to just full captions.

* The comparison with Streaming (Zhou et al., 2024) may not be completely fair as it focuses on memory management and not semantics.

* The approach relies on precomputed embeddings for the action phrases.

* Lines 110-111: "Unlike existing methods that use either text prefixing or embedding fusion..." lack concrete explanation.

* Lines 112-113: "We simplify action phrases into compact forms, such as action-object pairs..." need further elaboration on failure cases e.g. :

 1. Actions that can't be reduced to simple action-object pairs:
    "gradually stirring while slowly pouring in liquid",
    "adjusting equipment settings while monitoring readings"

 2. Actions requiring temporal context:
    "continuing to mix until consistency changes"


 3. Actions with multiple objects or relationships:
    "transferring contents from bowl to pan"
* I am not sure about the training stability here as switching between modes i.e. Action retrieval-augmented training and
standard training happens.

**Questions:**

- Why use CLIP embeddings vs other embedding approaches? Have you tried VLMs?
- What's the optimal trade-off between action phrase simplicity and informativeness? Can we include multiple actions and/or multiple objects?
- How sensitive is the model to segment length?
- What's the process to find the optimal ratio between augmented/non-augmented training? and How does this affect model convergence?
- What's the impact of action phrase quality? Have you done any study?
- What causes temporal misalignment? Can you provide some examples?

---

### Official Review · Reviewer_49JV · 2024-11-08

**Soundness:** 3
**Presentation:** 2
**Contribution:** 3
**Rating:** 5
**Confidence:** 3

**Summary:**

In this paper, the authors are trying to tackle a very challenging task, i.e., dense video captioning. One challenge in this task with recently proposed method is to effectively capture and utilize the evolving context in video streams. To improve this, an online retrieval-augmented approach, Actions Inspire Every Moment(AIEM), is proposed to process video segments incrementally. Additionally, an image-based simulated video pretraining strategy is introduced, where images are stitched to mimic video frames. The proposed framework achieves competitive results among all tested benchmarks.

**Strengths:**

Instead of using raw captions as previous methods, by incrementally retrieving action phrases that align temporally with each video segment, the proposed model achieves higher relevance and accuracy in captioning.

The model is evaluated on ViTT, YouCook2, and ActivityNet benchmarks and has consistent performance improvements over other baselines, including recently proposed online captioning framework.

Numerous ablation studies have been conducted with test details provided.

**Weaknesses:**

In my point of view, the paper is not well structured. Although the details of each step are provided, I found it a bit hard to follow the texts. It would be better if the key points could be focused and explained with some visual examples.

There are numerous ablation studies conducted, however, detailed analysis are missed. E.g. in 4.2.2, a key step, i.e., image-based simulated video pretraining, is tested. It indeed improves the performance, but the in-depth analysis is missed.

The proposed method requires a high-quality action-object pair which is a simple and effective way to improve the caption quality. I wonder, if it comes to complex interactions between multiple entities or multi-step actions, will the proposed framework can still be efficient.

**Questions:**

As aforementioned, I think the paper is in general not in a good form. the key components of the proposed method are somehow defocused with other details.

Please address the questions mentioned in the "Weaknesses" section.

---

### Note · Authors · 2024-11-15

I have read and agree with the venue's withdrawal policy on behalf of myself and my co-authors.